# CORN: Contact-based Object Representation for Nonprehensile Manipulation of General Unseen Objects

**Yoonyoung Cho,*  Junhyek Han,*  Yoontae Cho,  Beomjoon Kim**
Korea Advanced Institute of Science and Technology
{yoonyoung.cho, junhyek.han, yoontae.cho, beomjoon.kim}@kaist.ac.kr

## Abstract

Nonprehensile manipulation is essential for manipulating objects that are too thin, large, or otherwise ungraspable in the wild. To sidestep the difficulty of contact modeling in conventional modeling-based approaches, reinforcement learning (RL) has recently emerged as a promising alternative. However, previous RL approaches either lack the ability to generalize over diverse object shapes, or use simple action primitives that limit the diversity of robot motions. Furthermore, using RL over diverse object geometry is challenging due to the high cost of training a policy that takes in high-dimensional sensory inputs. We propose a novel contact-based object representation and pretraining pipeline to tackle this. To enable massively parallel training, we leverage a lightweight patch-based transformer architecture for our encoder that processes point clouds, thus scaling our training across thousands of environments. Compared to learning from scratch, or other shape representation baselines, our representation facilitates both time- and data-efficient learning. We validate the efficacy of our overall system by zero-shot transferring the trained policy to novel real-world objects. Code and videos are available at https://sites.google.com/view/contact-non-prehensile.

## 1 Introduction

Robot manipulators can transform our day-to-day lives by alleviating humans from taxing physical labor. Despite this potential, they have remained largely confined to picking and placing objects in limited settings such as manufacturing or logistics facilities. One key reason for such confined usage is their limited action repertoire: while prehensile manipulation has made great improvements (Mahler et al., 2017; Wan et al., 2023), there are several situations in the wild where pick-and-place is not an option. For instance, consider the scenarios in Figure 1, where the robot is tasked to manipulate diverse household objects to a target pose. While the effective strategy differs for each object, the predominant approach of grasping and placing objects is not viable. To manipulate such objects in the wild, the robot must perform nonprehensile actions such as pushing, dragging, and toppling by reasoning about the object geometry (Kao et al., 2016).

Traditionally, this problem has been tackled using planning algorithms that model the dynamics of the system and compute motions using optimization (Posa et al., 2014) or tree search (Cheng et al., 2022). However, this necessitates full system information about object geometry and inertial parameters, which are generally unavailable from sensory observations, especially for novel objects. Even when such models are available, the complex search space renders planning algorithms impractically slow. Recently, deep reinforcement learning (RL) has surfaced as a promising alternative, where a nonprehensile policy is trained in a simulator (Kim et al., 2023; Zhou et al., 2023), then zero-shot transferred to the real world. However, no algorithm offers the level of generality over object shapes and diversity in motions necessary for our problem: they either generalize over objects but are confined to rudimentary motions such as poking (Zhou et al., 2023) or affords rich motions but uses a rudimentary object representation that precludes generalization over shapes (Kim et al., 2023).

---

*equal contribution

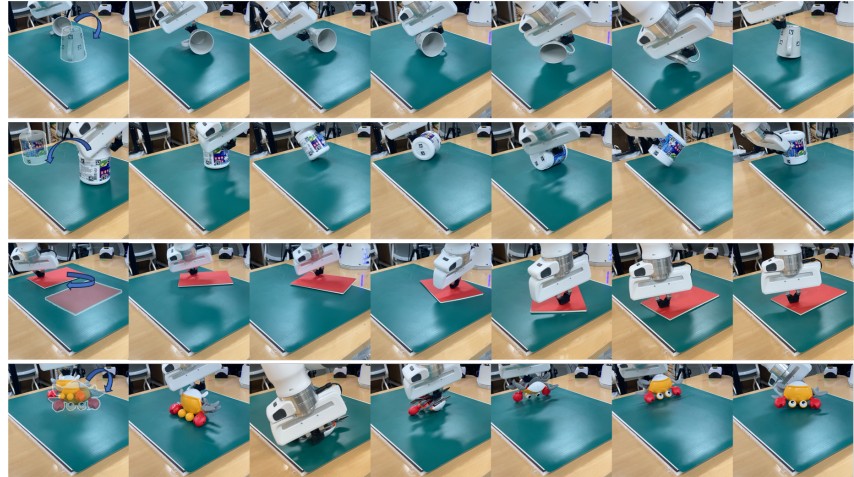

Figure 1: Snapshot of our robot performing diverse real-world object manipulation tasks. The first column shows the initial and goal object pose (transparent) of the task, and the green region marks the robot's workspace. **Row 1**: Raising a cup bottom-side up. The cup's grasp and placement poses are occluded by the collision with the table; since the cup is dynamically unstable, the robot must also support the object to avoid toppling the object out of the workspace during manipulation. **Row 2**: Flipping the wipe dispenser upside down, which is too wide to grasp. Since it may roll erratically, frequent re-contacts and dense closed-loop motions are required to stabilize the object during manipulation. **Row 3**: Moving a book that is too thin to be grasped; to drag the book, the robot must finely adjust the pressure to allow for both reorientation and translation. **Row 4**: Flipping a toy crab. Given its concave geometry, the robot must utilize the interior contacts to pivot the crab.

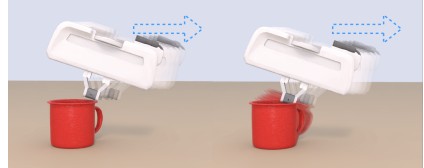

(a) Case where the robot tries to topple the cup.   (b) Case where the robot tries to push the block.

Figure 2: Even among similar-looking states, the interaction outcome varies drastically depending on the presence of contact. (a-left) the gripper passes near the cup, yet not quite in contact. As the robot narrowly misses the object, the object remains still. (a-right) the gripper engages with the cup, leading to a successful topple. (b-left) The robot relocates to the left of the block to push it to the goal(dark). By avoiding unintended collision, it is well-positioned to push the object. (b-right) due to spurious contact, the gripper accidentally topples the block, making the goal farther to reach.

We propose a novel object representation learning algorithm for nonprehensile manipulation that can be jointly used with the action-space from Kim et al. (2023) to achieve both generality over shapes and diversity in motions. The key challenge is defining the pretraining task. Given that our objective is to generalize across shapes, one naive approach is to employ self-supervised shape-learning, where the geometric representation is trained on contrastive or shape-reconstruction objectives (Pang et al., 2022; Yu et al., 2022; Xie et al., 2020; Zeid et al., 2023). However, reconstructing the full shape representation is needlessly complex: the full shape includes many intricate features, such as internal geometry, concavities, or surface details, which may not directly influence the performance of the manipulation policy. Even for a complex object, only a subset of its surface interacts with the gripper. Thus, while the contactable parts of an object must be represented at high fidelity, the remainder can be omitted without influencing the performance of the downstream policy.

Our key insight is that recognizing the presence and location of contacts between the object and the robot's hand is crucial for nonprehensile manipulation. For RL, a good representation must necessarily distinguish between states with high and low values, and in contact-rich manipulation, this is primarily determined by what forces and torques can be applied to the object in the given state.

This, in turn, depends on the presence and position of the contact between the object and the robot's hand. This intuition is demonstrated in Figure 2. Based on this insight, we propose a pretraining task where the point cloud encoder is trained to predict what part of the object is in collision with the robot's end-effector .

Our other contribution is the design of an efficient architecture that can process high-dimensional point clouds, in a way that can scale to massively parallel GPU-backed simulations. Existing well-known point-cloud processing architectures such as PointNet (Qi et al., 2016; 2017) inherently incur substantial inference time and memory usage while processing individual points. Inspired by the recent success of patch-based vision architectures (Dosovitskiy et al., 2020; Yu et al., 2022; Pang et al., 2022; Zhang et al., 2022; Zeid et al., 2023), we leverage a patch-based transformer to efficiently encode the point cloud. Since a single point in a point cloud does not carry significant information, grouping local points into a single unit(patch) (Yu et al., 2022) can effectively reduce the computational burden of launching thousands of point-wise operations as in PointNets (Qi et al., 2016) or Point-based Transformers (Zhao et al., 2021), resulting in significant computational gains.

We call our framework **C**ontact-based **O**bject **R**epresentation for **N**onprehensile manipulation (CORN). We show that by leveraging CORN, we can efficiently train a nonprehensile manipulation policy that can execute dexterous closed-loop joint-space maneuvers without being restricted to predefined motion primitives. Further, by employing an efficient patch-based point-cloud processing backbone, we can achieve highly time-efficient training by allowing the policy to scale to massively parallel environments. By adopting our full system, we demonstrate state-of-the-art performance in nonprehensile manipulation of general objects in simulated environments and zero-shot transfer to unseen real-world objects.

## 2 RELATED WORK

**Planning algorithms** In planning-based approaches, a nonprehensile manipulation problem is often tackled using gradient-based optimization or graph search. In optimization, the primary challenge arises from optimizing the system of discontinuous dynamics, stemming from contact mode transitions. To combat this, prior works resort to softening contact mode decision variables (Mordatch et al., 2012) or introducing complementarity constraints (Posa et al., 2014; Moura et al., 2022). However, due to the imprecision from the smooth approximation of contact mode transitions, inexact contact dynamics models, and difficulty in precisely satisfying contact constraints, the resulting motions are difficult to realize in the real world. On the other hand, graph-search approaches handle discontinuous system dynamics by formulating the problem as a search over a graph on nodes encoding states such as the robot's configuration and the object's contact mode, and the edges encoding transition feasibility and relative motion between the nodes (Maeda & Arai, 2005; Maeda et al., 2001; Miyazawa et al., 2005). As a result of handling discrete dynamics transitions without approximation, these methods can output more physically realistic motions that can be deployed in the real world (Cheng et al., 2022; Liang et al., 2022). However, to make the search tractable, they resort to strong simplifications such as quasi-static assumption on the space of motions (Cheng et al., 2022; Hou & Mason, 2019), or restricting the edges to predefined primitives or contact modes (Zito et al., 2012). As a result, these works have been limited to tasks that only require simple motions and sparse contact-mode transitions.

Moreover, because both of these approaches must compute a plan in a large hybrid search space with discontinuous dynamics, they are too slow to use in practice. Further, they also require the physical parameters of objects, such as mass and friction coefficients, to be known a priori, which undermines the practicality of these methods. Since the objects may vary significantly in the wild (Figure 5), acquiring ground-truth geometric and physical attributes of target objects is intractable.

**Reinforcement learning algorithms** Recently, several works have proposed to leverage learning to directly map sensory inputs to actions in nonprehensile manipulation tasks, circumventing the computational cost of planning, inaccuracy of hybrid dynamics models, and difficulty in system identification from high dimensional sensory inputs (Yuan et al., 2018; 2019; Lowrey et al., 2018; Peng et al., 2018; Ferrandis et al., 2023; Kim et al., 2023; Zhou & Held, 2023; Zhou et al., 2023). In Kim et al. (2023), they propose an approach that outputs diverse motions and effectively performs time-varying hybrid force and position control (Bogdanovic et al., 2020), by using the end-effector target pose and controller gains as action space. However, since they represent object geometry via

its bounding box, their approach has a limited generalization capability across shapes. Similarly, other approaches only handle simple shapes such as cuboids (Yuan et al., 2018; 2019; Ferrandis et al., 2023; Zhou & Held, 2023), pucks (Peng et al., 2018), or cylinders (Lowrey et al., 2018).

In HACMan (Zhou et al., 2023), they propose an approach that generalizes to diverse object shapes using end-to-end training. However, they are limited to 3D push primitives defined as fixed-direction poking motions applied on a point on the point cloud. Due to this, they must retract the arm after each execution to observe the next potential contact locations. Further, as end-to-end training with point clouds requires a significant memory footprint, it is difficult to use with massively parallel GPU-based simulations (Makoviychuk et al., 2021): the large resource consumption significantly limits the number of environments that can be simulated in parallel, undermining the parallelism and damaging the training time. Moreover, end-to-end RL on high-dimensional inputs is prone to instability (Eysenbach et al., 2022) and sample-inefficiency: noisy gradients from RL serve as poor training signals for the representation model (Banino et al., 2022), and distributional shift that occurs whenever the encoder gets updated (Shah & Kumar, 2021) slows down policy training. We believe for these reasons, they had to resort to simple primitives that make exploration easier. We use pretraining and efficient architecture to get around these problems.

Along with HACMan, other methods use simple primitives (Yuan et al., 2019; Zhou et al., 2023). But for problems we are contemplating, use of such open-loop primitives precludes dense feedback motions. This is problematic for highly unstable (e.g. rolling or pivoting) objects that move erratically, which require the robot to continuously adjust the contact. For instance, in Figure 1 (row 1, column 3), the robot must adjust the contact to prevent spurious pivoting of the ceramic cup while rolling the handle to the other side; in Figure 1 (row 2, column 6), the robot must align the contact with the rotational axis while lifting the wipe dispenser to prevent accidentally toppling the dispenser to either side. In our work, we adopt a controller with end-effector subgoals with variable gains (Bogdanovic et al., 2020; Kim et al., 2023) as our action space, which allows us to perform dense, closed-loop control of the object.

**Representation learning on point clouds** Diverse pretraining tasks have been proposed for learning a representation for point cloud inputs, such as PointContrast (Xie et al., 2020), which uses a contrastive loss on point correspondences, or OcCo (Wang et al., 2021), which reconstructs occluded points from partial views. More recently, inspired by advances from NLP (Devlin et al., 2018) and 2D vision (He et al., 2022; Baevski et al., 2022), self-supervised representation learning methods on masked transformers have gained attention (Yu et al., 2022; Zhang et al., 2022; Pang et al., 2022; Zeid et al., 2023). While Point-BERT (Yu et al., 2022), PointM2AE (Zhang et al., 2022) and PointMAE (Pang et al., 2022) reconstruct the masked points from the remaining points, Point2Vec (Zeid et al., 2023) follows Data2Vec (Baevski et al., 2022) and instead estimate the latent patch embeddings, outperforming above baselines in shape classification and segmentation.

As a result, these methods learn highly performant shape representations, demonstrating state-of-the-art performance across tasks such as shape classification or segmentation (Zeid et al., 2023). However, the same impressive performance is often inaccessible with smaller models (Fang et al., 2021; Shi et al., 2022) due to the difficulty of the pretraining task (Fang et al., 2021) or model collapse (Shi et al., 2022). While this necessitates large models, adopting a large encoder for policy training in GPU-based simulators is undesirable, as it can dominate the compute-cost and memory footprint during policy rollout, limiting the parallelism in massively-parallel RL setups.

In robotics, several alternative representation learning approaches have been proposed. Huang et al. (2021) pretrains their point cloud encoder by predicting object class and relative offset from the goal orientation. Similarly, Chen et al. (2022) pretrains a sparse 3D CNN on object class, relative orientation from the goal, and the joint positions of the robot hand. While these methods have been proven effective on in-hand re-orientation tasks for inferring the object's pose from point cloud observations, the benefits are smaller for nonprehensile manipulation tasks which are more sensitive to the specific presence and location of contacts (see Figure 2)[1]. In comparison, we directly represent an object with an embedding that is used to predict such information.

---

[1]While (Chen et al., 2022) demonstrated that in-hand reorientation task can be solved solely from pose-based observations without observing the underlying object geometry, the same success cannot be achieved in our domain (Figure 6)

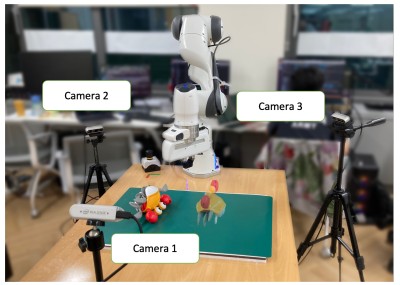
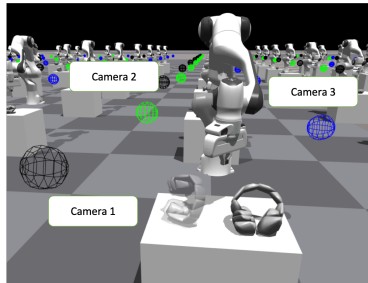

(a) Real-world setup.                    (b) Simulation setup.

Figure 3: Our real-world (left) and simulated (right) domains.

# 3 CONTACT-BASED OBJECT REPRESENTATION FOR NON-PREHENSILE MANIPULATION

We tackle the problem of rearranging an object of arbitrary geometry on a tabletop to a specified 6D relative pose from the initial pose using nonprehensile manipulation, such as pushing, toppling, and pivoting. Our setup includes a manipulator with a simple gripper, equipped with a proprioceptive sensor and table-mounted depth cameras. We assume that a rigid-body target object is well-singulated on the table, and both the initial and goal poses are stable at rest. Figure 3 illustrates our environment setup.

## 3.1 OVERVIEW

Our framework goes through three stages of training before being deployed in the real world: (1) pretraining, (2) teacher policy training with privileged information, and (3) student policy training with partial information only available in the real world. These three stages happen entirely in simulation based on Isaac Gym (Makoviychuk et al., 2021). Our overall system is shown in Figure 4.

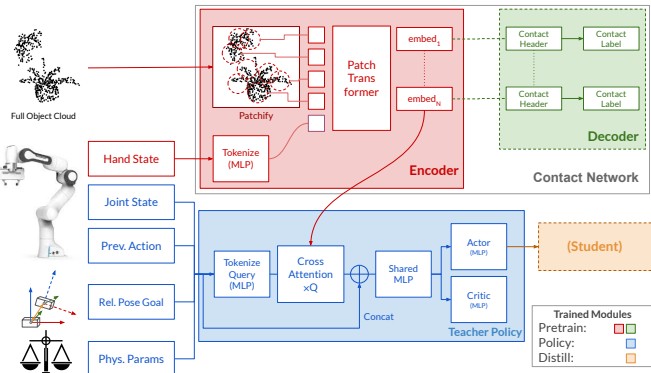

Figure 4: Our system and model architecture. The contact network consists of a point cloud encoder (red) and contact-prediction decoder (green), passing the point cloud embeddings to the teacher policy module (blue). Student module (orange, omitted) is detailed in Section A.1.

In the pretraining phase, we train our encoder module (Figure 4, red), which takes in point-cloud observations $\in \mathbb{R}^{512 \times 3}$ and the hand state $\in \mathbb{R}^9$ (position and 6D orientation (Zhou et al., 2019)), and outputs a patch-wise embedding of the object cloud. To train the network, we pass the embeddings through a contact prediction header(Figure 4, green) that classifies whether each patch intersects with the gripper. During this phase, the point-cloud encoder and collision decoder are trained jointly.

During policy training, we discard the collision decoder and freeze the encoder, so that only the weights of the policy module are learned. The policy consumes the object point cloud embedding from the encoder, and other task-relevant inputs comprising robot's joint position and velocity, the

previous action (end-effector subgoal and controller gains), the object's pose relative to the goal, and its physical properties (mass, friction, and restitution). These inputs are tokenized via an MLP layer to produce query tokens for the cross-attention layer against the point cloud embeddings to extract task-relevant features. These features are concatenated again with the task-relevant inputs and processed by a shared MLP before being split into actor and critic networks. We train the policy module via PPO (Schulman et al., 2017). Since full-cloud observations and physics parameters of objects are not available in the real world, we distill the teacher policy into a student policy. We use DAgger (Ross et al., 2011) to imitate the teacher's actions solely based on information available in the real world: the partial point cloud of the object, robot's joint states, the previous action, and the relative object pose from the goal. We utilize nvdiffrast (Laine et al., 2020) to efficiently render the partial point cloud of the scene on the GPU during training.

## 3.2 Learning CORN

Algorithm 1 outlines the data generation procedure for our network, inspired by (Son & Kim, 2023). We first sample the SE(3) transforms within the workspace for the gripper and object (L1-2). We then compute the nearest displacement between the surface points of the object and the gripper (L3) and move the gripper in that direction plus a small Gaussian noise (L4-6) to approach a configuration near or in a collision. To compute the labels, we sample points from the object's surface (L7), then label each point according to whether they intersect with the gripper (L8-10). This process generates about half of the gripper poses in collision with the object, and the other half in a near-collision pose with the object. To disambiguate these cases, the model must precisely reason about the object's geometry.

---

**Algorithm 1** Dataset generation for CORN.

---

**Require:** object shape $\mathcal{O}$, gripper shape $\mathcal{G}$, workspace $\mathcal{W}$, noise level $\sigma$
**Ensure:** gripper pose $T_G$, object cloud $X$, contact label $L$
1: $T_O, T_G \leftarrow SamplePoses(\mathcal{W})$
2: $\mathcal{O}' \leftarrow T_O \cdot \mathcal{O}; \mathcal{G}' \leftarrow T_G \cdot \mathcal{G}$
3: $\vec{\delta} \leftarrow NearestDisplacement(\mathcal{O}', \mathcal{G}')$
4: $s \sim \mathcal{N}(1, \sigma/||\vec{\delta}||)$
5: $T_d \leftarrow \begin{bmatrix} I_{3\times3} & -s \cdot \vec{\delta} \\ 0 & 1 \end{bmatrix}$
6: $\mathcal{G}'' \leftarrow T_d \cdot \mathcal{G}'$
7: $X \sim SampleSurfacePoints(\mathcal{O}')$
8: **for** $x_i \in X$ **do**
9: $\quad L_i \leftarrow PointInMesh(x_i, \mathcal{G}'')$
10: **end for**

---

Afterward, the model is trained by classifying whether each patch of the object intersects with the gripper at the given pose. A patch is labeled as positive if any point within it intersects with the gripper. We use binary cross-entropy between the patchwise decoder logits and the collision labels to train the model(Figure 4, green block)

## 3.3 Patch-based architecture for CORN

To efficiently train a policy by simultaneously simulating thousands of environments using GPUs, we leverage a patch-based transformer to encode the object point cloud shown in Figure 4 (red), which pass the patch embeddings to the policy module (blue).

**Patch Tokenization.** In line with previous work (Pang et al., 2022), we divide the point cloud into a set of patches (Figure 4, red block, top left). To define each patch, we identify $N$ representative points using farthest-point sampling (FPS). The k-nearest neighbor (kNN) points from these representatives constitute a patch, which is normalized by subtracting its center from its constituent points (Pang et al., 2022; Zeid et al., 2023; Yu et al., 2022) to embed the local shape. These patches are then tokenized via a Multi-Layer Perceptron (MLP) as input tokens to a transformer.

Unlike past methods that use a small PointNet for tokenizing the patches (Pang et al., 2022; Zeid et al., 2023), we tokenize the patches with a lightweight MLP. Using an MLP is sufficient since the patch size is constant, fully determined by the size of $k$ during kNN lookup. By sorting the points within each patch by their distance from the patch center, we can ensure that points within a patch always follow a consistent order. Thus, we can avoid using a costly permutation-invariant PointNet tokenizer. After computing the local patch embeddings, we add learnable positional embeddings of the patch centers to restore the global position information (Pang et al., 2022).

**Additional Encoder Inputs.** By leveraging a transformer backbone, our architecture can easily mix heterogeneous task-related inputs as extra tokens, such as hand state, together with raw point-cloud observations as shown in Figure 4. Unlike PointNet-derived models that can only concatenate such inputs *after* the point-processing layers, this has an advantage in that the network can pay attention to parts of the point cloud in accordance with the hand state, such as the ones near the robot's hand that would interact with the gripper.

## 4 EXPERIMENTS

### 4.1 EXPERIMENTAL SETUP

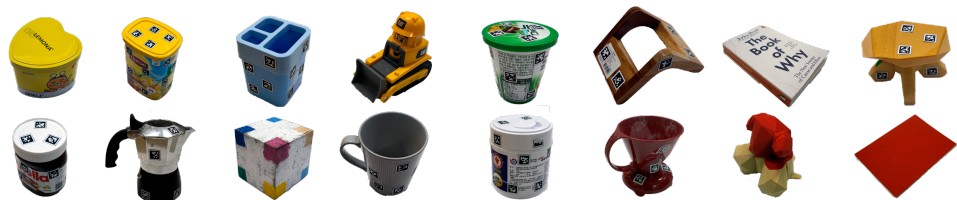

Figure 5: Set of 16 real-world objects that we test in the real world.

**Training setup** For each scene, we randomly draw an object from a subset of 323 geometrically diverse objects from DexGraspNet dataset (Wang et al., 2023), detailed in Section A.6.2. For each episode, we first place the object in a random stable pose on the table. Then, we reset the robot arm at a joint configuration uniformly sampled within predefined joint bounds slightly above the workspace. By doing so, we can avoid initializing the robot in spurious collision with the table or the object. Afterward, we sample a 6D stable goal pose randomly on the table, spaced at least 0.1m away from the initial pose to prevent immediately achieving success at initialization. To sample valid initial and goal poses for each object, we pre-compute a set of stable poses for each object. This procedure is elaborated in Section A.6.1.

**Action, Rewards, and Success Criteria** Following Kim et al. (2023), the action space of the robot comprises the subgoal residual of the hand, and the joint-space gains of a variable impedance controller (Details in Section A.2). We also follow the same curriculum learning scheme on the subgoal residual from Kim et al. (2023) to facilitate transferring our policy to the real world. The reward in our domain is defined as $r = r_{suc} + r_{reach} + r_{contact} - c_{energy}$, comprising the task success reward $r_{suc}$, goal-reaching reward $r_{reach}$, contact-inducing reward $r_{contact}$, and energy penalty $c_{energy}$. The task success reward $r_{suc} = \mathbb{1}_{suc}$ is given when the pose of the object is within 0.05m and 0.1 radians of the target pose. We add dense rewards $r_{reach}$ and $r_{contact}$ to facilitate learning, based on a potential function (Ng et al., 1999) and has the form $r = \gamma \phi(s') - \phi(s)$ where $\gamma \in [0, 1)$ is the discount factor. For $r_{reach}$, we have $\phi_{reach}(s) = k_g \gamma^{k_d \cdot d_{o,g}(s)}$ and for $r_{contact}$, we have $\phi_{contact}(s) = k_r \gamma^{k_d \cdot d_{h,o}(s)}$ where $k_g, k_d, k_r \in \mathbb{R}$ are scaling coefficients; $d_{o,g}(s)$ is the distance from the current object pose to the goal pose, measured using the bounding-box based distance measure from Allshire et al. (2022); $d_{h,o}(s)$ is the hand-object distance between the object's center of mass and the tip of the gripper. Energy penalty term is defined as $c_{energy} = k_e \cdot \sum_{i=1}^{7} \tau_i \cdot \dot{q}_i$, where $k_e \in \mathbb{R}$ is a scaling coefficient, $\tau_i$ and $\dot{q}_i$ are joint torque and velocity of the $i^{th}$ joint. The scaling coefficients for the rewards are included in Table A.1.

**Domain Randomization.** During training, we randomize the mass, scale of the object, and friction and restitution of the object, table, and the robot gripper. We set the scale of the object by setting the largest diameter of the object within a predefined range. To facilitate real-world transfer, we also add a small noise to the torque command, object point cloud, and the goal pose when training the student. Detailed parameters for domain randomization are described in Table A.4.

**Real-World Setup.** Our real-world setup is shown in Figure 3a. We use three RealSense D435 cameras to extract the point clouds, and use Iterative Closest Point (ICP) To estimate the error between the object's current and target poses. We further use April tags (Wang & Olson, 2016) to track symmetric objects that cannot be readily tracked via ICP. We also wrap the gripper of the

robot with a high-friction glove to match the simulation. We emphasize that all training is done in simulation, and the policy is zero-shot transferred to the real world.

## 4.2 RESULTS

We'd like to evaluate the following claims: (1) Pretraining CORN enables training a nonprehensile manipulation policy in a time- and data-efficient manner. (2) Patch-based transformer architecture enables efficiently scaling to thousands of point-cloud based RL agents in parallel. (3) Our system can generalize to domain shifts such as unseen objects in the real world. To support claims 1 and 2, we compare our pretraining task and architecture (OURS) against the baseline encoders in Figure 6.

Compared to ours, P2V uses a pretrained Point2Vec (Zeid et al., 2023) encoder, using the author's weights; P2V-LITE is pretrained as in P2V, but uses the same architecture as OURS; ROT also shares the architecture with OURS, but is pretrained with relative rotation estimation as in (Huang et al., 2021; Chen et al., 2022); E2E also shares the architecture, but is trained in an end-to-end manner without pretraining; PN uses a three-layer PointNet encoder, pretrained on point-wise collision prediction, analogous to CORN; and STATE trains a geometry-unaware policy that only receives object bounding box inputs as the object representation, rather than point clouds. Note that P2V, P2V-LITE, ROT, E2E have the same downstream policy architectures as ours. For PN and STATE, they also have the same policy architecture except that they do not use cross-attention.

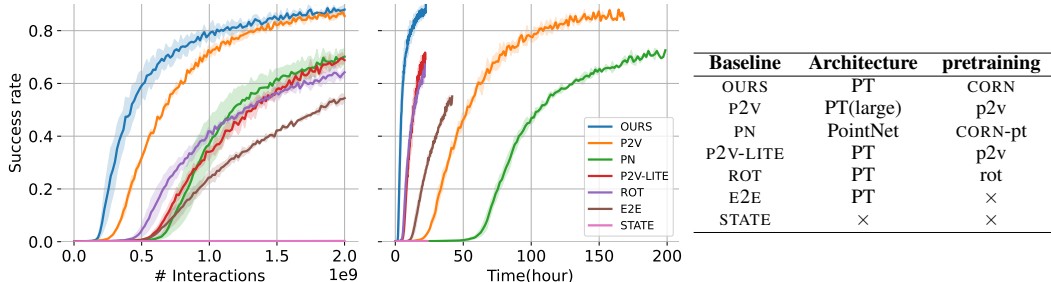

Figure 6: Training progression and baselines. Plots show the mean (solid) and standard deviations (transparent) for each baseline. (Left) Success rate vs. number of interaction steps. Interaction steps are reported as the total number of steps aggregated across 4096 parallel environments. (Middle) Success rate vs. number of hours. All baselines are allowed to interact at most $2e9$ time steps. (Right) Comparison of baselines. PT: Patch Transformer.

Figure 6 shows the progression of success rates across different baselines in simulation. To support claim 1, we consider the performance of the policy after 2 billion interactions (in steps) and 24 hours (in time), respectively. First, E2E takes significantly longer compared to OURS, in terms of both steps and time: by pretraining CORN, OURS achieves 88.0% suc.rate (steps) and 88.5% (time), while E2E only reaches 31.4% (in time) and 54.4% (in steps), respectively. This is due to the overhead of jointly training the encoder and policy networks in E2E. Among pretrained representations, we first compare with P2V. While P2V is similar to OURS in steps(85.6%), it is significantly slower in time(3.55%), due to the inference overhead of the large representation model. To isolate the effect of the pretraining task, we also compare with P2V-LITE: pretrained as in P2V, yet with same architecture as OURS. While the resulting policy is reasonably time-efficient due to faster rollouts (73.4%, in time), it is less data-efficient (68.8%, in steps) due to the reduced performance of the smaller model (Shi et al., 2022; Fang et al., 2021). Another baseline, ROT from in-hand reorientation literature (Huang et al., 2021; Chen et al., 2022) performs slightly worse in both measures(64.2% (steps); 67.0% (time)): we believe this is due to the increased demand for the awareness of object geometry in our task. To see this, we note that the pose-only policy(STATE), unaware of object geometry, cannot learn at all in our domain, in contrast to the positive result from Chen et al. (2022). This highlights the fact that our task requires greater awareness of the object's geometric features: just knowing the object's pose is insufficient, and a deeper understanding of its geometry is necessary.

To see CORN learns a suitable representation for manipulation, we inspect the attention scores of the policy amid different motions in Figure 7. For pushing and toppling motions (row 1,3), our policy primarily attends to the current or future contact between the hand and the object, which is consistent

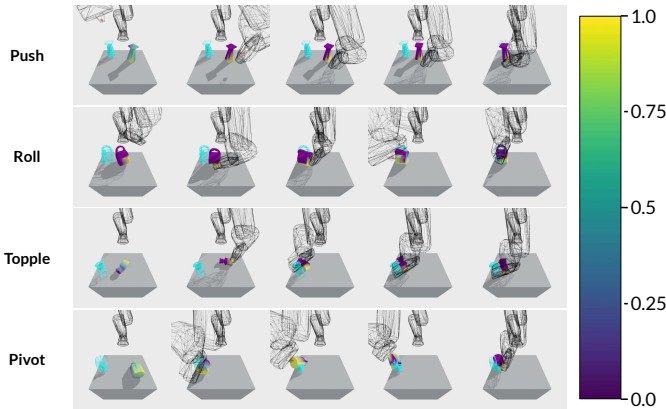

Figure 7: Visualization of the scores from the cross-attention layers of the policy, summed over all heads. We colorize the attention for each patch normalized to range $(0, 1)$, then project the patch-wise colors on the surface of the object for visualization with VIRIDIS colormap.

with the premise of our representation-learning scheme. While pivoting or rolling motions (row 2,4) share the same trend, our policy also attends to the pivot points of the object against the table.

To support claim 2, we show that our patch-transformer architecture is more performant in both compute- and memory-cost compared to PointNet (Qi et al., 2016). In particular, we evaluate PN, where the primary difference is the use of PointNet in place of the patch transformer. While the performance degradation from the architecture change is only modest in steps (88.3% v. 70.2%), the large computational overhead of PointNet renders it slowest in time(0.223%) among all baselines. To support claim 3, we evaluate our policy in the real-world setting as described in Section 4.1. Figure 5 shows the objects that we evaluate in our real-world experiments, exhibiting wide diversity in terms of shape, mass, and materials; Figure 1 shows four representative real-world episodes.

Table 1: Real world run results. All of the objects in the real-world setup are unseen except for 3D-printed objects marked with †.

| Object name | Success/Trial | Object name | Success/Trial | Object name | Success/Trial | Object name | Success/Trial |
|---|---|---|---|---|---|---|---|
| Red Card | 4/5 | Cube Block | 4/5 | Ceramic Cup | 4/5 | Red Dripper | 3/5 |
| Toy Puppy† | 4/5 | Iced Tea Box | 3/5 | Nutella Bottle | 3/5 | Coaster Holder | 3/5 |
| Toy Bulldozer | 4/5 | Wipe Dispenser | 5/5 | Coffee Jar | 2/5 | Thick Book | 4/5 |
| Blue Holder | 5/5 | Toy Table† | 3/5 | Lemona Box | 3/5 | Candy Box | 3/5 |
| Success rate | | | | | | | 57/80 (71.3%) |

The results from our policy execution in the real-world are organized in Table 1. Overall, our policy demonstrates 71.3% success rate in the real world. These results indicate that, with distillation, our system can zero-shot transfer to the real world despite only training in a simulation.

**Limitations.** We find that our policy struggles more with concave objects (coaster-holder, red dripper; 60%) or unstable objects (iced tea box, coffee jar, red dripper; 53%). Concave objects cause perception challenge in the real world due to severe ambiguity when occluded by the robot, leading to timeouts as the agent oscillates between actions or stops amid a maneuver. Unstable objects tend to abruptly spin out of the workspace after minor collisions, and the dexterity of our robot platform and speed of our real-world perception pipeline cannot catch up with such rapid motions.

## 5 CONCLUSION

In this work, we present a system for effectively training a non-prehensile object manipulation policy that can generalize over diverse object shapes. We show that by pretraining a contact-informed representation of the object, coupled with an efficient patch-based transformer architecture, we can train the policy in a data- and time-efficient manner. With student-teacher distillation, we find that our policy can zero-shot transfer to the real world, achieving success rates of 71.3% in the real world across both seen and unseen objects despite only training in a simulation.

## ACKNOWLEDGEMENT

This work was supported by Institute of Information & communications Technology Planning & Evaluation (IITP) grant funded by the Korea government(MSIT) (No.2019-0-00075, Artificial Intelligence Graduate School Program(KAIST)), (No.2022-0-00311, Development of Goal-Oriented Reinforcement Learning Techniques for Contact-Rich Robotic Manipulation of Everyday Objects), (No. 2022-0-00612, Geometric and Physical Commonsense Reasoning based Behavior Intelligence for Embodied AI).

## REPRODUCIBILITY STATEMENT

We describe our real-world system and perception pipeline in detail in Section A.4. The procedure for data generation and pretraining for our model and other baselines is detailed in Section 3.2 and Section A.3, respectively. We include the set of 323 training objects from DexGraspNet (Wang et al., 2023) that we used for policy training as supplementary materials. We include the student-model architecture and distillation pipeline for real-world transfer in Section A.1. We describe the simulation setup regarding data preprocessing(Section A.6), and describe the hyperparameters associated with our network architecture(Table A.2), domain-randomization(Table A.4), and policy training(Section A.3). We also release our code at `https://github.com/contact-non-prehensile/corn`.

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

# A APPENDIX

## A.1 STUDENT MODEL

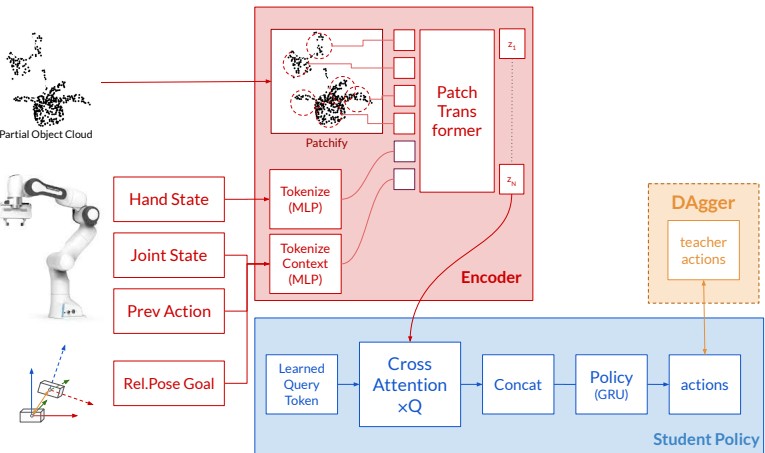

Figure A.1: Student Model Architecture in DAgger.

The student policy shares a similar architecture with the teacher policy, with two distinctions. Unlike teacher policy, which employs joint state, previous action, and goal inputs for generating query tokens for cross-attention, we incorporate these tokens into the input of the patch transformer within the encoder to facilitate extracting task-relevant features from partial point clouds. Given that these inputs are already incorporated in the input of the encoder, we use learned query tokens for cross-attention.

Since the student policy can only perceive limited shape information of objects at any given moment due to occlusion, we incorporate a two-layer Gated Recurrent Unit (GRU) (Cho et al., 2014) on top of the student point cloud encoder. This enables the student model to aggregate partial observations of object geometry based on historical observations.

We train the student with DAgger with truncated backpropagation through time (BPTT). In particular, we run the student for 8 time-steps and update its parameters to minimize KL-divergence between the action distribution of the teacher policy and the student policy.

## A.2 DETAILS ON POLICY ACTION SPACE

The action space of our policy consists of the subgoal residual of the end-effector $\Delta T_{ee}$ and joint-space gains. The subgoal residual is parameterized by the translational offset $\Delta t \in \mathbb{R}^3$ and axis-angle representation of the residual rotation in the world frame $\Delta r \in \mathbb{R}^3$. The joint-space gains are parameterized by proportional terms ($k_p \in \mathbb{R}^7$) and damping factors ($\rho \in \mathbb{R}^7$) of a variable impedance controller. Based on this, we compute the joint position target by solving inverse kinematics with damped least squares method (Buss, 2004) as $q_{target} = q_t + IK(\Delta T_{ee})$. The torque command for each joint is computed based on the joint position error and the predicted joint-space gains and damping factors as $\tau = k_p(q_{target} - q_t) - k_d \dot{q}_t, k_d = \rho \cdot \sqrt{k_p}$.

## A.3 DATA GENERATION AND PRETRAINING PIPELINES

**P2V-LITE.** For pretraining P2V-LITE, we follow the same procedure as Zeid et al. (2023), except we use the same architecture as OURS. Due to the use of a significantly lighter model, we apply two modifications to the original training setup, as follows: first, since we only use a two-layer transformer, instead of averaging over the last 6 layers of the teacher, we take the result after layer-normalization of the final layer, which keeps the ratio consistent (50% of layers). Second, we replace DropPath layers with dropout since the model is shallow. Otherwise, we follow the original hyper-

parameters from the paper and the author's code(`https://github.com/kabouzeid/point2vec`).

**ROT.** For pretraining ROT, we follow the procedure described in Chen et al. (2022). Since the authors report that adding the classification objective does not influence the encoder quality, we omit the classification terms. For the training data, we use ShapeNet Chang et al. (2015) dataset. We apply two different SO(3) transformations to the same point cloud, and pass it through the encoder to obtain the patchwise embeddings. We concatenate the patchwise embeddings from both streams, then pass it through a 3-layer MLP (512, 64, 6). We train the encoder on the regression task, where the task for the decoder is to regress the parameters for a 6D rotation Zhou et al. (2019) as recommended from Chen et al. (2022).

**OURS-PN.** For pretraining PN, we use the same dataset from OURS. Since PointNet does not produce patchwise embeddings, we instead adopt a U-Net architecture based on PyG Fey & Lenssen (2019). To inform the model about the end-effector, we concatenate the hand pose (position and orientation Zhou et al. (2019)) to each point in the channel dimension.

## A.4 REAL WORLD SYSTEM AND PERCEPTION PIPELINE

**Point Cloud Segmentation.** In the point cloud segmentation phase, we integrate three RealSense D435 cameras. Points outside the predefined 3D bounding-box workspace above table are first removed, and those within a distance less than the threshold $\epsilon = 1\,\text{cm}$ from the table surface are removed next. These parameters were determined empirically to handle outliers without removing too much of the object. For thin objects such as cards, we alternatively employ color-based segmentation to remove the table from the point cloud, avoiding the risk of inadvertently removing object points.

After removing the table, only the object and the robot remains. To remove the robot, we compute the transform of all robot links based on the robot kinematics, then query the points as to whether it is contained within each link mesh. For efficiency, we first apply convex decomposition on the robot geometry, and employ point-convex intersection to check whether the point belongs to the robot. We initially tried using robot-points removal based on raycasting the point-to-robot mesh distances, but we found it was much slower than the convex-hull based method.

To reduce noise in the point cloud, we employ a radius outlier removal logic with $r = 2$ cm and $n_{\min} = 96$ points, which is the minimum number of points that the inlier should contain within the neighborhood. Finally, to identify the object point cluster, we used DBSCAN clustering algorithm. Clusters were computed with $\epsilon$=1cm and the number of neighbors as 4 to form the core-set.

To enhance processing speed, we optimized the entire segmentation pipeline to utilize tensor operations executed on the GPU. The output object point cloud of this segmentation process serves as input for both our policy and object pose tracking pipeline.

**Object Pose Tracking.** To trade-off tracking speed and performance, we first subsample the object point clouds randomly to $n_{\text{track}} = 2048$ points. Empirically, we found that this avoids compromising the quality of the ICP(Iterative Closest Point) algorithm, while remaining reasonably fast.

We employ ICP for tracking. Given the object point cloud $C_t$ at current time step $t$, we conduct point-to-plane ICP registrations with point cloud of previous timestep $C_{t-1}$, resulting in a transformation $P_{t,t-1} = \text{ICP}(C_t, C_{t-1})$. The current object pose $T_{O_t}$ is obtained by recursively applying pairwise transformations: $T_{O_t} = P_{t,t-1} \cdot T_{O_{t-1}}$. In the case that the transform drifts over time, we also perform an additional ICP match with the initial object point cloud $P_{t,0} = \text{ICP}(C_t, C_0)$ to correct the error. If the fitness score $Fitness_{t,0}$ exceeds 60%, current object pose is calculated as $T_{O_t} = P_{t,0} \cdot T_{O_0}$.

For thin objects, we utilize point-to-point ICP matching since their point cloud nearly forms a single plane. While the ICP tracker performs well for large and non-symmetrical objects, it struggles with highly symmetrical objects, or objects with point-matching ambiguity under occlusion (e.g. when cup handles are occluded by the robot).

To address this challenge, we tried color-based ICP to resolve the ambiguity based on surface texture. However, we found that it did not improve tracking performance and, in some cases, even

worsened it. As such, we used april-tag tracker to maintain object pose tracking accuracy for highly symmetrical objects.

## A.5 EXAMPLE TRAJECTORIES

Example trajectories can be observed from the video in the supplementary material, or from `http s://sites.google.com/view/contact-non-prehensile`.

## A.6 SIMULATION SETUP

### A.6.1 STABLE POSES GENERATION

To sample stable poses for training, we drop the objects in a simulation and extract the poses after stabilization. In 80% of the trials, we drop the object 0.2m above the table in a uniformly random orientation. In the remaining 20%, we drop the objects from their canonical orientations, to increase the chance of landing in standing poses, as they are less likely to occur naturally. If the object remains stationary for 10 consecutive timesteps, and its center of mass projects within the support polygon of the object, then we consider the pose to be stable. We repeat this process to collect at least 128 initial candidates for each object, then keep the unique orientations by pruning equivalent orientations that only differ by a planar rotation about the z-axis.

### A.6.2 OBJECTS FOR TRAINING

We sample 323 objects from the DexGraspNet dataset (Wang et al., 2023), shown in Figure A.2. To increase the simulation speed and stability, we apply convex decomposition on every mesh using CoACD (Wei et al., 2022), after generating the watertight mesh with simplification using Manifold (Huang et al., 2018).

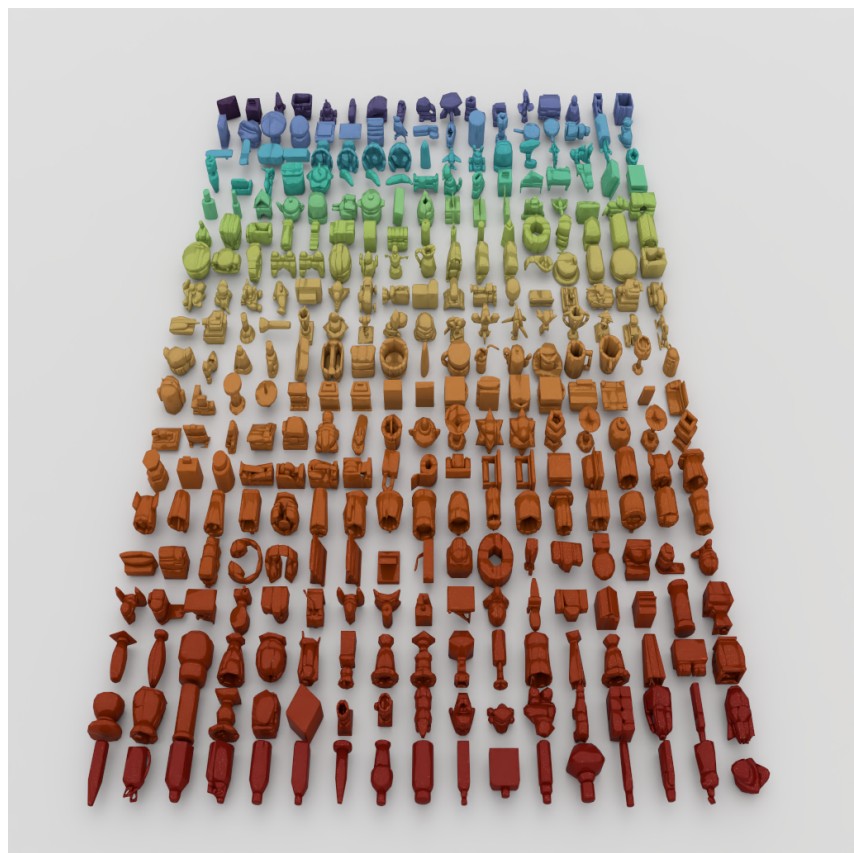

Figure A.2: 323 object meshes that were used for training.

## A.7 HYPERPARAMETERS

Table A.1: Hyperparmeters for reward computation.

| Parameter | Value | Description |
|---|---|---|
| $k_g$ | 0.302 | Goal-reaching reward coefficient |
| $k_r$ | 0.0604 | Hand-reaching reward coefficient |
| $k_e$ | 0.0001 | Energy penalty coefficient |
| $k_d$ | 243.12 | Decay factor for potential-based reward |

Table A.2: Hyperparameters for Encoder and Policy.

| Hyperparameter | Value | Hyperparameter | Value | Hyperparameter | Value |
|---|---|---|---|---|---|
| Num. points | 512 | Hidden dim. | 128 | Shared MLP | MLP (512, 256, 128) |
| Num. patches | 16 | Num. layer | 2 | Actor | MLP (64, 1) |
| Patch size | 32 | Num. attention head in self-attention | 4 | Critic | MLP(64, 20) |
| Concatenate context input | True | Num. attention head in cross-attention (teacher/student) | 16/4 | | |

Table A.3: Hyperparameters for PPO.

| Hyperparameter | Value | Hyperparameter | Value | Hyperparameter | Value | Hyperparameter | Value |
|---|---|---|---|---|---|---|---|
| Max Num. epoch | 8 | Base learning rate | 0.0003 | GAE parameter | 0.95 | Num. environment | 4096 |
| Early-stopping KL target | 0.024 | Adaptive-LR KL target | 0.016 | Discount factor | 0.99 | Episode length | 300 |
| Entropy regularization | 0 | Learning rate schedule | adaptive | PPO clip range | 0.3 | Update frequency | 8 |
| Policy loss coeff. | 2 | Value loss coeff. | 0.5 | Bound loss coeff. | 0.02 | | |

Table A.4: Range for domain randomization. $\mathcal{U}[min, max]$ denotes uniform distribution, and $\mathcal{N}[\mu, \sigma]$ denotes Normal distribution. Point cloud noise and Goal pose noise are added to the normalized input.

| Parameter | Range |
|---|---|
| Object mass (kg) | $\mathcal{U}[0.1, 0.5]$ |
| Object scale (m) | $\mathcal{U}[0.1, 0.3]$ |
| Object friction | $\mathcal{U}[0.7, 1.0]$ |
| Table friction | $\mathcal{U}[0.3, 0.8]$ |
| Gripper friction | $\mathcal{U}[1.0, 1.5]$ |
| Torque noise | $\mathcal{N}[0.0, 0.03]$ |
| Point cloud noise | $\mathcal{N}[0.0, 0.005]$ |
| Goal pose noise | $\mathcal{N}[0.0, 0.005]$ |

