# OpenReview forum: "CORN: Contact-based Object Representation for Nonprehensile Manipulation of General Unseen Objects"
_ICLR.cc/2024/Conference — ICLR 2024 poster_

### Official Review · Reviewer_UC4D · 2023-10-27

**Soundness:** 3 good
**Presentation:** 2 fair
**Contribution:** 2 fair
**Rating:** 8
**Confidence:** 4

**Summary:**

This paper presents an innovative contact-based representation for non-prehensile manipulators, which aims to enhance the robot's ability to manipulate objects. A pre-training model is utilized to predict the contact between the gripper and the object, thereby providing the policy with a more detailed understanding of the robot-object interaction. The state-based policy is then distilled into a vision-based one for implementation in real-world scenarios.

**Strengths:**

1. The creation of this representation is commendable, as it underscores the importance of using the full functionality of the gripper for contact. The algorithm designed for its training is intriguing and appears to be well thought out.
2. The experimental results convincingly demonstrate the model's capability, efficiency, and superiority compared to alternative methods. Furthermore, experiments using real robots with zero-shot transfer highlight the model's robustness, thereby solidifying the research.

**Weaknesses:**

1. The scope of manipulation tasks in this research is restricted to single-object state maneuvering using a closed gripper. Incorporating more intricate robot-object interactions, such as grasping, could help to fully utilize the robot's kinematic capabilities and cover more complex scenarios.
2. The paper measures the success of the tasks using a "success rate." However, necessary details, like the specific criteria used to determine successful manipulation, lacks clarity. Moreover, the goal state illustrated in the manipulation videos (or their screenshots) appears to be a snapshot of a future state, which does not accurately represent the actual desired outcome.

**Questions:**

1. Could you elaborate on the "success rate" metric used in your experiments? What specific criteria are used to determine a successful manipulation?
2. Can you explain how do you measure the physical parameters in the real world and make sure they are aligned or well-simulated in simulations?

---

> ### Author Response · Authors · 2023-11-17
> **Response to Reviewer UC4D**
>
> Thank you for your thoughtful comments and suggestions. We addressed your comments, and would like to provide additional details regarding your questions as follows:
>
> > (W1) The scope of manipulation tasks in this research is restricted to single-object state maneuvering using a closed gripper. Incorporating more intricate robot-object interactions, such as grasping, could help to fully utilize the robot's kinematic capabilities and cover more complex scenarios.
>
> While we agree that our scope involves manipulating a single object using a closed gripper, as far as we know, this is the first time that anyone has gotten a real robot to do non-prehensile manipulation that utilizes full 6D end-effector pose on the diversity of objects that we had. Most of the existing methods are limited to either simple primitives (e.g. poking objects) [1] or are limited to objects for which they know the physical parameters (mass, friction coefficients, center of mass) [2].
>
> Also, grasping using a simple gripper ***does not*** involve as intricate robot-object interactions as non-prehensile manipulation — it simply involves reaching the object, and then closing the gripper, and has been studied significantly in the past [3].  In contrast, because non-prehensile manipulation requires careful choice of contacts, mode switches, and joint movements to maneuver the object, it is known to be a more difficult problem [4].
>
> > (W2-1) The paper measures the success of the tasks using a "success rate." However, necessary details, like the specific criteria used to determine successful manipulation, lacks clarity.
> >
> > (Q1) Could you elaborate on the "success rate" metric used in your experiments? What specific criteria are used to determine a successful manipulation?
>
> We describe task success criteria in section 4.1 as the translational offset of the pose within 0.05m distance and angular 0.1 radians. We updated the draft to better elucidate this point in the experiment section.
>
> > (W2-2) Moreover, the goal state illustrated in the manipulation videos (or their screenshots) appears to be a snapshot of a future state, which does not accurately represent the actual desired outcome.
>
> The goal visualized in the video ***is*** the approximation of the object at the actual desired pose. Obtaining goal images of objects after positioning objects precisely at the goal pose is impractical; we believe this is a reasonable visualization. Below is the actual difference in average pose error between the final object pose and its desired goal location in the real-world experiments.
>
> | Metric | Value |
> | --- | --- |
> | Translation Error (m) | 0.036 ± 0.010 |
> | Orientation Error (rad) | 0.068 ± 0.019 |
>
> > (Q2) Can you explain how do you measure the physical parameters in the real world and make sure they are aligned or well-simulated in simulations?
>
> Actually, we don’t need to measure the physical parameters. The purpose of distillation (section 3.1) is to remove the need for privileged information, such as the physical parameters. We distill our teacher policy (section 3.1) so that the student policy can operate in the real world without the physical parameters as input. The student architecture, detailed in A.1, also shows that it does not require physical parameters as input.
>
> [1] Zhou, Wenxuan, et al. "HACMan: Learning Hybrid Actor-Critic Maps for 6D Non-Prehensile Manipulation." *7th Annual Conference on Robot Learning (CoRL)*. 2023.
>
> [2] Cheng, Xianyi, et al. "Contact mode guided motion planning for quasidynamic dexterous manipulation in 3d." *2022 International Conference on Robotics and Automation (ICRA)*. IEEE, 2022
>
> [3] Sundermeyer, Martin, et al. "Contact-graspnet: Efficient 6-dof grasp generation in cluttered scenes." *2021 IEEE International Conference on Robotics and Automation (ICRA)*. IEEE, 2021.
>
> [4] Lynch, Kevin Michael. *Nonprehensile robotic manipulation: Controllability and planning*. Carnegie Mellon University, 1996.

---

> > ### Comment · Reviewer_UC4D · 2023-11-22
> > **Response to the Rebuttal**
> >
> > Thanks to the authors for their efforts. Most of my problems and concerns have been addressed, for which I will raise my score.

---

### Official Review · Reviewer_dbmK · 2023-10-30

**Soundness:** 2 fair
**Presentation:** 2 fair
**Contribution:** 2 fair
**Rating:** 5
**Confidence:** 4

**Summary:**

The paper presents a novel method for nonprehensile manipulation using reinforcement learning (RL). Traditional RL struggles with diverse object shapes and high-dimensional sensory inputs. The authors introduce a contact-based object representation and pretraining pipeline, using a patch-based transformer encoder for point clouds, enabling scalable training. Their approach offers time- and data-efficient learning, with successful zero-shot transfers to real-world objects.

**Strengths:**

1. The paper motivates well, and the techniques presented are sound.
2. The paper is well-written and easy to follow.
3. The presented method shows good generalization ability to unseen objects and sim-to-real scenarios.

**Weaknesses:**

1. The novelty of this work is marginal. basically, it just applies point-cloud-based reinforcement learning to nonprehensile manipulation tasks.

2. The generation of the collision label is questionable with the coverage of the collision states, i.e., if this work truly aims for generalization on unseen objects, I wonder how the collision prediction network be generalized to the unseen geometry.

3. Besides, if the contact network is trained as a guide for the encoder, how does it guarantee to generate a collision-free policy? The collision decoder itself is not perfect (due to the coverage issue and neural network prediction), and the influence on the encoder is indirect (leads to less perfect), not to mention the policy is distilled from the teacher network to the student network (even less perfect).

4. Since this method is "contact-based"? In a broad sense, all the manipulation is contact-based, but I assume the word choice here is for the embedding induced by the contact network. Then, though the authors have conducted extensive baseline methods on the encoder side, in my opinion, they should also discuss different decoder schemes to justify the "contact-based" name. For example:

(a). the encoder is directly trained by the policy network? That is, no contact decoder to pretrain the encoder.

(b). the decoder is a geometry reconstruction network. In this sense, the encoder can capture the geometry information of the object point cloud and hand. I see no apparent reason why the "action and rewards" cannot be done with such kinds of decoders.

5. Another possibility of the naming is due to the r_contact in the reward, but the contact potential is not given exactly, since the essential d_h,o is built upon the distance from the CoM of the object to the tip. Which does not prevent collision/intersection between the gripper and hand.

Therefore, if the contact is not necessarily determinant or explicitly modeled for policy training in both network design and reward shaping, the only sane way to interpret the contact-based would be in the broadest sense, i.e., all the manipulation is contact-based. In this sense, I would suggest removing the "contact-based" in the title, which would give wrong expectations to the readers. Or the paper can present stronger relevance of "contact" to the object representation.

**Questions:**

See the weakness section above.

---

> ### Author Response · Authors · 2023-11-17
> **Response to Reviewer dbmK**
>
> We appreciate the reviewer’s effort in providing the review. Unfortunately, the reviewer seems to have a major misunderstanding about our paper and raised concerns based on this misunderstanding.
>
> First, our goal is not to learn a “collision-free policy” that computes a collision-free motion. As the title of our paper and Figure 1 on page 2 suggest, our problem is that of non-prehensile manipulation, and object manipulation cannot be done without making contact with an object (We suggest the reviewer to checkout the video of our robot performing the task of interest: [https://sites.google.com/view/contact-non-prehensile](https://sites.google.com/view/contact-non-prehensile)). In particular, our problem involves learning where to contact the object and how to apply forces to an object, which involves deliberately colliding with the object. The key aspect of this problem, therefore, is knowing where the robot has contacted the object because that tells us what forces the robot can apply to the object. Our contribution is the design of pretext task and neural network architecture tailored to detect such contacts.
>
> The reviewer raised two concerns based on this misunderstanding. The first was the naming of our method (it has “contact-based” in it). We hope the reasoning above tells you why contact-based representation is important.
>
> The second is whether our contact predictor generalizes to novel objects, because even a small mistake would make the robot collide with the object. We hope the reviewer now has a better understanding of the task, and how this would not be an issue.
>
> > (W1) The novelty of this work is marginal. basically, it just applies point-cloud-based reinforcement learning to non-prehensile manipulation tasks.
> - While there have been approaches that consider point-cloud-based RL in robot manipulation, no prior work has solved closed-loop non-prehensile manipulation of general objects. The inherent complexity in contact dynamics between the object, robot and the environment in non-prehensile manipulation renders naive RL application to be prohibitively expensive in our domain without nontrivial design choices, as seen in our results (Figure 7). As such, no prior work has demonstrated the generalization over diverse object geometries with closed-loop feedback control in this domain.
>
> > (W2) The generation of the collision label is questionable with the coverage of the collision states, i.e., if this work truly aims for generalization on unseen objects, I wonder how the collision prediction network be generalized to the unseen geometry. Besides, if the contact network is trained as a guide for the encoder, how does it guarantee to generate a collision-free policy? The collision decoder itself is not perfect (due to the coverage issue and neural network prediction), and the influence on the encoder is indirect (leads to less perfect), not to mention the policy is distilled from the teacher network to the student network (even less perfect).
>
> > Can collision network generalize to unseen object geometry? How can we ensure that the policy can precisely avoid collision despite errors coming from factors like (indirect guidance, limited coverage and approximation error from distillation)?
> - Yes. Our network generalizes to unseen geometry, as evident in our experimental results. All except one of real-world objects are unseen, and the simulation test setting involves novel objects as well. We kindly encourage the reviewer to give a read of our experiment section.
>
> > (W3) Then, though the authors have conducted extensive baseline methods on the encoder side, in my opinion, they should also discuss different decoder schemes to justify the "contact-based" name. For example:
> >
> > (a). the encoder is directly trained by the policy network? That is, no contact decoder to pretrain the encoder.
> >
> > (b). the decoder is a geometry reconstruction network. In this sense, the encoder can capture the geometry information of the object point cloud and hand. I see no apparent reason why the "action and rewards" cannot be done with such kinds of decoders.
>
> - The baselines you mentioned have already been implemented in the experiment section.
>     - In Section 4.2, the End-to-end (E2E) baseline is the one that the encoder is directly trained by the policy network i.e. RL objective. This corresponds to option (a) from your recommendation.
>     - In Section 4.2, the Point2Vec (P2V) baseline represents the pretraining scheme where the encoder is trained on shape-reconstruction objectives. Point2Vec learns shape representation by reconstructing the latent shape embeddings (similar to data2vec[1]). This corresponds to option (b) from your recommendation.
>     - Ours outperform these both in terms of data and computational efficiency. (Figure 7)
>
> [1] Baevski, Alexei, et al. "Data2vec: A general framework for self-supervised learning in speech, vision and language." *International Conference on Machine Learning*. PMLR, 2022

---

> > ### Comment · Reviewer_dbmK · 2023-11-21
> > **Reply**
> >
> > Let me explain why I think the contact-based is not proper, or at least not clear.
> >
> > First, contact is a kind of state when the robot gripper/hand touches the surface of the objects. It is not a state where the gripper is away from the object's surface or inside the object's surface. We can distinguish these states with the term "collision". "contact" is surely in "collision", but "collision" is not "contact". "contact" is an intermediate state between "in-collision" and "collision-free".
> > In my view, the author misuses these two terms. I suggest the author refer to Springer Handbook of Robotics, Contact Modeling and Manipulation, and check related sections. (https://link.springer.com/referenceworkentry/10.1007/978-3-540-30301-5_28)
> >
> > In the domain of robotics, the term "contact" has a special meaning. It refers to the contact interface. If the authors only use the word "contact" as a normal verb, as they did at the end of Sec. 3.2, "A patch is labeled as 'collision' if any point within it contacts the gripper", and use contact/collision interchangeably, I can accept that, but please make it clear.
> >
> > I have a feeling it is the same reason why reviewer LafH thinks this term usage is problematic. But I will only speak for myself.
> >
> > Another way to interpret this is that the contact surface is indirectly determined by in-collision/collision-free, just like how SDF indirectly defines the surface by inside/outside indicators.
> > In this sense, it will raise the issue of generalization to unseen geometry, which happens to SDF-based 3D reconstruction tasks.
> > Of course, I noticed the claim of generalization to unseen geometry in the main paper, but the objects used are quite similar to trained objects. I just don't think it will generalize to a larger object set (it already does not perform well on slightly complex objects, Table 1).
> >
> > As for the collision-free policy. First, obviously, there will never be an intersection between a gripper and an object in the real world. Thus, the discussion on in-collision/collision-free is based on the training, which takes place in simulation. My question is, how is the collision in simulation avoided when the contact reward is defined on the distance between the object's CoM to the gripper tip, not the object's surface? Will the possible collision make the object move unexpectedly? Could the bad performance on concave objects (irregular place of CoM) and unstable objects (changing CoM) be because of such contact reward design?
> > I assume the interpenetration is handled by the collision detection methods implemented natively in the simulator. If so, the contact reward is just a usable reward, not a meticulous design.
> >
> > Finally, let's look at the technical contribution. I think the core contribution here is the learning of hand-object features and applying them to teacher-student learning. The problem is, there's really no necessity of this hand-object representation to non-prehensile manipulation tasks. The learning framework can perfectly work with grasping tasks by slightly changing the reward function.
> >
> > Thus, I will think of this work as good engineering work for robotics. The indirect contact modeling can be written more clearly, and the contact reward can be designed with more thought.
> >
> > It is a half-half decision here, but I lean to retain my initial rating.

---

> ### Author Response · Authors · 2023-11-23
> **Response to Reviewer dbmK (1/3)**
>
> > Let me explain why I think the contact-based is not proper, or at least not clear.
> >
> > First, contact is a kind of state when the robot gripper/hand touches the surface of the objects. It is not a state where the gripper is away from the object's surface or inside the object's surface. We can distinguish these states with the term "collision". "contact" is surely in "collision", but "collision" is not "contact". "contact" is an intermediate state between "in-collision" and "collision-free". In my view, the author misuses these two terms. I suggest the author refer to Springer Handbook of Robotics, Contact Modeling and Manipulation, and check related sections. ([https://link.springer.com/referenceworkentry/10.1007/978-3-540-30301-5_28](https://link.springer.com/referenceworkentry/10.1007/978-3-540-30301-5_28))
> >
> > In the domain of robotics, the term "contact" has a special meaning. It refers to the contact interface. If the authors only use the word "contact" as a normal verb, as they did at the end of Sec. 3.2, "A patch is labeled as 'collision' if any point within it contacts the gripper", and use contact/collision interchangeably, I can accept that, but please make it clear.
> >
> > I have a feeling it is the same reason why reviewer LafH thinks this term usage is problematic. But I will only speak for myself.
>
> * The main point raised by the reviewer here is to disambiguate the use of “contact” (for signed distance d == 0) and “in-collision” (d≤0). We agree that this disambiguation is important in collision-free motion planning, where the penetration depth is used to define the cost function for trajectory optimization, for instance. We use the two terms interchangeably in our setup, since there’s little point in discriminating the two conditions in our task.
> * As we train an RL agent, the training process occurs primarily through robot-object interaction. In real-world interactions, penetration between objects cannot occur; and in a simulation, the momentary penetrations are handled by the physics engine itself during contact resolution. Thus, from the perspective of the agent, there’s no need to disambiguate the two conditions (d==0, d≤0).
> * That being said, we appreciate the reviewer’s suggestions on clarity. We have updated Section 3.2 to communicate that the pre-training labels primarily consider penetration between objects.
>
> > Another way to interpret this is that the contact surface is indirectly determined by in-collision/collision-free, just like how SDF indirectly defines the surface by inside/outside indicators. In this sense, it will raise the issue of generalization to unseen geometry, which happens to SDF-based 3D reconstruction tasks. Of course, I noticed the claim of generalization to unseen geometry in the main paper, but the objects used are quite similar to trained objects. I just don't think it will generalize to a larger object set (it already does not perform well on slightly complex objects, Table 1).
>
> * We appreciate the reviewer’s concern that SDF-based 3D reconstruction tasks often suffer from generalization. Of course, it’s only natural that representations learned through neural nets would be bound to generalize within the distribution of objects similar to the training set. However, we’d like to clarify that our objective is to predict the collision between *local* patches and the robot gripper, rather than the much harder task of predicting the *global* implicit shape. This type of *locality* is often utilized in prior works[1,2] as a key ingredient to generalizing across diverse shapes, as the geometry of local parts tends to be quite commonly shared across different objects in a combinatorial manner, thus boosting the generalization over different shapes.
> * While the reviewer wonders whether the performance drop on “complex” objects is due to the difficulty of generalizing to objects of complex shapes — which could be partly true — we believe that this has more to do with the severe occlusion and dynamic instability of objects rather than the difficulty of our representation to generalize to complex geometry, as discussed in Limitations (Section 4.2).

---

> > ### Author Response · Authors · 2023-11-23
> > **Response to Reviewer dbmK (2/3)**
> >
> > > As for the collision-free policy. First, obviously, there will never be an intersection between a gripper and an object in the real world. Thus, the discussion on in-collision/collision-free is based on the training, which takes place in simulation. My question is, how is the collision in simulation avoided when the contact reward is defined on the distance between the object's CoM to the gripper tip, not the object's surface? Will the possible collision make the object move unexpectedly? Could the bad performance on concave objects (irregular place of CoM) and unstable objects (changing CoM) be because of such contact reward design? I assume the interpenetration is handled by the collision detection methods implemented natively in the simulator. If so, the contact reward is just a usable reward, not a meticulous design.
> >
> > - We want to note that the purpose of the contact-inducing reward $r_{contact}$ is not to prevent collision, but to guide initial exploration. This strategy is commonly adopted in contact-rich manipulation tasks [3,4] to facilitate policy training: during initial exploration, the robot is prone to make meaningless actions (i.e. move in thin air) without making significant progress. To prevent this, the contact-inducing reward encourages the policy to reduce the distance between the gripper and the object, hence increasing the probability of the robot interacting with the object during initial exploration.
> > - How does the policy learn to avoid spurious collision that might result in undesired motion of the object? Since the policy is primarily driven to maximize task success (which is the dominant reward), and the contact-inducing reward just serves as guidance, the policy is incentivized to avoid collision if it reduces the likelihood of success.
> > - As for the specific implementation of the contact-inducing reward, we’d like to note that during our early investigation, we also tried the surface-to-surface distance (rather than CoM distance) based contact reward but this did not make a significant difference in policy training. This is only natural, as approximately guiding the robot towards the general region of the object is sufficient to encourage interaction during initial exploration, which is the purpose of the contact-inducing reward.
> >
> > > Finally, let's look at the technical contribution. I think the core contribution here is the learning of hand-object features and applying them to teacher-student learning. The problem is, there's really no necessity of this hand-object representation to non-prehensile manipulation tasks. The learning framework can perfectly work with grasping tasks by slightly changing the reward function.
> >
> > The point raised by the reviewer is worded ambiguously, so we respond to two possible interpretations of the statement.
> >
> > (Interpretation 1) There’s no reason to *limit* this hand-object representation to non-prehensile manipulation tasks; it seems it could work equally as well for prehensile manipulation.
> >
> > - While our representation can also be technically applied to grasping, we believe that the main benefit is on non-prehensile manipulation tasks. The primary purpose of adopting representation learning is to stabilize and expedite the process of training an RL policy, since jointly learning the representation of high-dimensional data and the behavior tends to be unstable [5,6].
> > - On the other hand, a grasp policy is typically directly learned via supervised training by leveraging large-scale offline datasets, such as ACRONYM [7] or Graspnet-1billion [8]. As such, learning an object representation a priori is typically not necessary in grasping.
> >
> > (Interpretation 2) There's no necessity to use this hand-object representation on non-prehensile manipulation tasks. *Even without this representation*, the policy can be trained just like grasping tasks by slightly changing the reward function.
> >
> > - We show that training with our representation drastically improves the training efficiency (Figure 7), which is otherwise prohibitively expensive. While it’s perhaps true that tuning the reward function may improve the training efficiency by some amount, we’d like to emphasize that it is quite crucial to adopt the proper representation, which results in an order-of-magnitude speedup in terms of training time.
> > - While the reviewer suggests that slightly changing the reward function may suffice for learning (as proposed in earlier comments), we comment on this in our discussion above. During our early investigations, modifying the contact-inducing reward to more closely model the true inter-object distance (as suggested by the reviewer), has yielded similar training performance. Rather than the specific choice of the reward, we believe that the proper choice of representation is quite central in our domain.
> >
> > We’re not sure which of the two interpretations is consistent with the reviewer’s intent, but hope that either response addresses the point raised by the reviewer.

---

> > > ### Author Response · Authors · 2023-11-23
> > > **Response to Reviewer dbmK (3/3)**
> > >
> > > [1] Jiang, Chiyu, et al. "Local implicit grid representations for 3d scenes." *Proceedings of the IEEE/CVF Conference on Computer Vision and Pattern Recognition*. 2020.
> > >
> > > [2] Dongwon Son and Beomjoon Kim. “Local object crop collision network for efficient simulation of non-convex objects in GPU-based simulators.” In Proceedings of Robotics: Science and Systems. 2023.
> > >
> > > [3] Chen, Tao, et al. "Visual dexterity: In-hand dexterous manipulation from depth." *ICML workshop on new frontiers in learning, control, and dynamical systems*. 2023.
> > >
> > > [4] Allshire, Arthur, et al. "Transferring dexterous manipulation from gpu simulation to a remote real-world trifinger." *2022 IEEE/RSJ International Conference on Intelligent Robots and Systems (IROS)*. 2022.
> > >
> > > [5] Shah, Rutav M., and Vikash Kumar. "RRL: Resnet as representation for Reinforcement Learning." *International Conference on Machine Learning*. PMLR, 2021.
> > >
> > > [6] Banino, Andrea, et al. Coberl: Contrastive bert for reinforcement learning. In *International Conference on Learning Representations*, 2022.
> > >
> > > [7] Eppner, Clemens, Arsalan Mousavian, and Dieter Fox. "Acronym: A large-scale grasp dataset based on simulation." *2021 IEEE International Conference on Robotics and Automation (ICRA).* 2021.
> > >
> > > [8] Fang, Hao-Shu, et al. "Graspnet-1billion: A large-scale benchmark for general object grasping." *Proceedings of the IEEE/CVF conference on computer vision and pattern recognition*. 2020.

---

### Official Review · Reviewer_LafH · 2023-10-31

**Soundness:** 2 fair
**Presentation:** 3 good
**Contribution:** 2 fair
**Rating:** 5
**Confidence:** 3

**Summary:**

The paper introduces CORN, a Contact-based Object Representation for Nonprehensile Manipulation of General Unseen Objects. The system combines deep reinforcement learning with a novel contact-based object representation to effectively manipulate a variety of shapes and sizes of objects. The key innovation lies in the use of a lightweight patch-based Transformer architecture to process point clouds, enabling large-scale parallel training across thousands of environments. The efficacy of CORN is validated through a series of experiments, demonstrating zero-shot transferability to novel real-world objects.

**Strengths:**

- The paper introduces an approach to nonprehensile manipulation of general unseen objects, a challenging area in robotics.
- The methodology is laid out with a clear structure, and the paper provides a set of experimental results to back its proposals.
- The work has potential implications for the field of robotics, particularly in object manipulation, although the full extent of its impact may require further exploration and validation.

**Weaknesses:**

- Lack of Unique Design for Complex Operations: The paper emphasizes in the introduction that its approach can execute more complex robotic arm operations than prior grasping work. However, in the method description, there appears to be no clear unique design to directly support this motivation, raising doubts about the novelty and effectiveness of the method.
- Overemphasis on "Contact": While the authors place particular emphasis on the importance of "contact" in nonprehensile operations, it seems that the equally vital role of "contact" in grasping tasks has not been adequately considered. This imbalanced emphasis may impact the comprehensiveness of the method and its practical applicability.
- Questioning the Novelty of Point Cloud Processing: The paper utilizes a Transformer-based architecture to process point cloud data, but this approach does not appear groundbreaking, emphasizing the need for a more detailed description of the policy network design.
- Insufficient Description of the Policy Network: The paper provides a relatively concise description of the policy network, lacking in-depth details, which might hinder readers from fully understanding the workings of the method and its potential advantages.
- Inadequate Experimental Setup: The paper does not offer detailed information related to actions and reward settings in the experiments, which is crucial for evaluating the effectiveness of the reinforcement learning portion.
In summary, these points highlight potential shortcomings in the paper concerning method description, evidence of novelty, and experimental setup. They offer specific directions for improvement to the authors, aiming to enhance the persuasiveness of the paper and provide a clearer conveyance of their research outcomes.

**Questions:**

- The paper strongly emphasizes the role of "contact" in nonprehensile manipulation. How does this emphasis differ in importance from its role in grasping tasks?
- The description of the policy network is relatively concise. Could the authors provide more details on its design and working principles?

---

> ### Author Response · Authors · 2023-11-17
> **Response to Reviewer LafH (1)**
>
> Thank you for your thoughtful comments and suggestions. We addressed your comments, and would like to provide additional details regarding your questions as follows:
>
> > (W1) Lack of Unique Design for Complex Operations: The paper emphasizes in the introduction that its approach can execute more complex robotic arm operations than prior grasping work. However, in the method description, there appears to be no clear unique design to directly support this motivation, raising doubts about the novelty and effectiveness of the method.
>
> - We’d like to highlight two unique design choices made to address the complexity of the non-prehensile manipulation task: the model architecture and object representation learning.
> - The complexity of contact dynamics of non-prehensile manipulation necessitates closed-loop feedback control. However, training such a dense, closed-loop policy is expensive in terms of both time and compute cost. To address this, we first adopt a patch-based transformer architecture, for the first time among works in point-cloud-based robot manipulation. This accelerates the training process and reduces the training time remarkably, as shown in Figure 7.
> - Secondly, we observe that non-prehensile manipulation is highly sensitive to the knowledge of contact between the object and the robot. To reflect this, we adopt a novel contact-based pretraining task. Together with our model architecture, we demonstrate significant efficiency gains as a result of adopting our pretraining scheme.
>
> > (W2) Overemphasis on "Contact": While the authors place particular emphasis on the importance of "contact" in nonprehensile operations, it seems that the equally vital role of "contact" in grasping tasks has not been adequately considered. This imbalanced emphasis may impact the comprehensiveness of the method and its practical applicability.
> >
> > (Q1) The paper strongly emphasizes the role of "contact" in nonprehensile manipulation. How does this emphasis differ in importance from its role in grasping tasks?
>
> - Non-prehensile manipulation requires careful choice of contacts, i.e. sequence of contact mode switches which defines what forces the robot can apply to the object. We strongly encourage the reviewer to check out the video of our method operating in the real world[[https://sites.google.com/view/contact-non-prehensile](https://sites.google.com/view/contact-non-prehensile/)]: this is in stark contrast with grasping with a parallel jaw gripper, which typically only involves approaching an object and closing the gripper.
>
> > (W3) Questioning the Novelty of Point Cloud Processing: The paper utilizes a Transformer-based architecture to process point cloud data, but this approach does not appear groundbreaking, emphasizing the need for a more detailed description of the policy network design.
>
> - Most prior works in robot manipulation actually only use PointNet-based architectures. We explore the choice of point cloud processing architecture, and show that patch-based transformer greatly improves computational efficiency in our domain, by about 8x in terms of model inference time and more than 20x in terms of policy training. To our knowledge, we are the first ones to demonstrate the efficacy of patch-based transformer in point-cloud based robot manipulation.
>
> > (W4) Insufficient Description of the Policy Network: The paper provides a relatively concise description of the policy network, lacking in-depth details, which might hinder readers from fully understanding the workings of the method and its potential advantages.
> >
> > (Q2) The description of the policy network is relatively concise. Could the authors provide more details on its design and working principles?
>
> - We apologize for missing the details, and updated the paper to include additional details. The policy starts with an initial cross-attention layer against the tokens of the current task context(joint state, previous action, goal, and physics parameters) to extract task-relevant information from the patch-level embeddings. The resulting features are concatenated again with the task context, which are then processed by an MLP(multi-layer perception) before being fed to actor and critic networks. Given that our focus is on the point-cloud processing representation model, we placed less emphasis on the policy network architecture when describing our model. We have now added extra details about the policy network to facilitate the reader’s understanding of our model. (Figure 5. & Section 3.1)

---

> > ### Author Response · Authors · 2023-11-17
> > **Response to Reviewer LafH (2)**
> >
> > > (W5) Inadequate Experimental Setup: The paper does not offer detailed information related to actions and reward settings in the experiments, which is crucial for evaluating the effectiveness of the reinforcement learning portion. In summary, these points highlight potential shortcomings in the paper concerning method description, evidence of novelty, and experimental setup. They offer specific directions for improvement to the authors, aiming to enhance the persuasiveness of the paper and provide a clearer conveyance of their research outcomes.
> >
> > - We appreciate the reviewer’s feedback regarding the necessity of additional details on the action-space. We have added a paragraph in the appendix that describes the structure of the action-space for the robot (Appendix A.2). However, we already provide a detailed breakdown of the reward function in section 4.1, and include relevant hyperparameters in Table A.1 (Appendix A.7), and we’d appreciate it if the reviewer could clarify which part of the reward function requires further clarification.

---

### Official Review · Reviewer_4HBK · 2023-11-06

**Soundness:** 4 excellent
**Presentation:** 4 excellent
**Contribution:** 4 excellent
**Rating:** 10
**Confidence:** 4

**Summary:**

This paper presents a novel method for non-prehensile manipulation, i.e. manipulating an object that is not graspable via poking, pivoting and toppling from an initial pose to a goal pose. There are two major technical contributions:
- A novel pre-training objective on predicting which parts of the object point cloud are in contact with the gripper
- A novel patch-based transformer architecture that allows efficient encoding of point clouds and other modalities such as robot gripper state.

The paper also provides a new dataset on non-prehensile manipulation with over 300 different objects.

**Strengths:**

- The model is trained entirely in simulation and achieves over 70% success rate in the real world with zero-shot sim-to-real transfer.
- The model finishes 2 million steps training in less than a day.
- The objects tested have good diversity. Over 20 objects are tested in the real world and over 300 objects are used in simulation. The test objects have significant geometric difference from the training objects.
- The paper addresses an important and difficult problem in contact-rich manipulation, which has significant impact on expanding a robot’s manipulation ability to more diverse objects in the real world.
- The experiments are very comprehensive, covering all the key parts of the design, including the point encoder and the pre-training objective.
- The hyperparameters are well documented in the appendix, which is important for reproducing the results.

**Weaknesses:**

- The model takes the difference between current object pose and target object pose as input. This can bring significant engineering challenge since object segmentation and pose tracking in the real world can be difficult. However, the authors documented their approach very well in Sec. A.3.
- The method only works on object well separated from clutter on a tabletop. This is related to the above assumption for the object pose, since having more than one object in close contact will make object segmentation and pose tracking even more challenging.

**Questions:**

- What is the coordinate frame of the hand pose input? Since the hand pose is sampled near the object to ensure good data balance for contact/no contact during pre-training, will the hand pose go out of distribution during policy learning?
- In Figure 7, is the scale of the success rate the same for the two plots? It seems that the final success rate doesn’t match up for the green, red, purple and brown curves.
- Which simulator is used? I didn’t find it in the paper.

---

> ### Author Response · Authors · 2023-11-17
> **Response to Reviewer 4HBK**
>
> We thank the reviewer for the effort and positive assessment of our work.
>
> > (W1) The model takes the difference between current object pose and target object pose as input. This can bring significant engineering challenge since object segmentation and pose tracking in the real world can be difficult. However, the authors documented their approach very well in Sec. A.3.
> >
> > (W2) The method only works on object well separated from clutter on a tabletop. This is related to the above assumption for the object pose, since having more than one object in close contact will make object segmentation and pose tracking even more challenging.
>
> We agree that segmentation and pose tracking are significant engineering challenges, and we appreciate the reviewer for recognizing the documentation of our approach. We are also hopeful that recent advances in foundational vision models, such as SAM[1] or POPE[2] can facilitate object pose tracking and segmentation to be more generally accessible for robot manipulation in the future, particularly for the challenges such as segmenting non-singulated objects.
>
> > (Q1) What is the coordinate frame of the hand pose input? Since the hand pose is sampled near the object to ensure good data balance for contact/no contact during pre-training, will the hand pose go out of distribution during policy learning?
>
> The hand pose is represented in the world frame. Since the poses are sampled from the workspace during pre-training, it does not actually go out of distribution during policy learning.
>
> > (Q2) In Figure 7, is the scale of the success rate the same for the two plots? It seems that the final success rate doesn’t match up for the green, red, purple and brown curves.
>
> There was a small bug in the plotting code, which has been now addressed. We updated Figure 7 and made minor adjustments on reported metrics in Section 4.2. Thank you for catching that!
>
> > (Q3) Which simulator is used? I didn’t find it in the paper.
>
> We used Isaac Gym. We added the reference to the simulation in Section 3.1.
>
> [1] Kirillov, Alexander, et al. "Segment anything.”, ****Proceedings of the IEEE/CVF International Conference on Computer Vision (ICCV), 2023, pp. 4015-4026
>
> [2] Fan, Zhiwen, et al. "POPE: 6-DoF Promptable Pose Estimation of Any Object, in Any Scene, with One Reference." *arXiv preprint arXiv:2305.15727* (2023).

---

### Comment · Area_Chair_pipJ · 2023-11-23
**Author-Reviewer discussion period ending *very* soon**

Thank you to the reviewers who have responded so far. The authors have put great effort into their response, so can I please urge reviewers **4HBK and LafH** to answer the rebuttal.
Thank you!

---

### Meta-Review · Area_Chair_pipJ · 2023-12-05

**Metareview:**

The paper introduces an approach using reinforcement learning (RL) for nonprehensile manipulation, crucial for handling thin or large objects. It addresses limitations of previous RL methods by proposing a contact-based object representation and pretraining method. Leveraging a patch-based transformer architecture, it enables efficient training across diverse object shapes and environments. Their approach shows improved efficiency in both time and data utilisation compared to other methods. The paper's success is demonstrated by transferring the trained policy to real-world objects without prior exposure.

Two reviewers were very positive of the paper, highlighting the following paper strengths: comprehensive experiments, intriguing algorithm design, and clarity of presentation. The other two reviewers were less positive; expressing scepticism regarding the novelty and efficacy of the proposed approach. They questions the generalisation of collision prediction to unseen geometries and highlights concerns about the indirect influence of the collision decoder on the encoder. Both reviewers challenged the interpretation of "contact-based" manipulation, advising adjustments to avoid misleading readers.

Overall, the paper reception was positive; however, this AC agrees that the paper would greatly benefit from clarifying the concept and motivation of "contacts", as suggested by reviewers LafH and dbmK.

**Justification For Why Not Higher Score:**

Although mostly positive, there were disagreements between reviewers on novelty/impact, and therefore may not be appropriate for a spotlight.

**Justification For Why Not Lower Score:**

Despite some questions around novelty from 2 of the reviewers, overall reviewers seemed to find the paper empirically strong. Moreover, one reviewer in particular was championing the paper (score of 10), with no reviewer pushing hard for the paper to be rejected.

---

### Decision · Program_Chairs · 2024-01-16

Accept (poster)